# Effect of Calcium on the Growth of Djulis (*Chenopodium formosanum* Koidz.) Sprouts

Yun-Yang Chao *[ID], Wei-Jia Wang and Yan-Ting Liu

Department of Plant Industry, National Pingtung University of Science and Technology. 1, Shuefu Road, Neipu, Pingtung 912301, Taiwan; ga36221212@gmail.com (W.-J.W.); b122830497@gmail.com (Y.-T.L.)
* Correspondence: chaoyy@mail.npust.edu.tw; Tel.: +886-8-7703202

**Abstract:** Taiwanese quinoa (*Chenopodium formosanum* Koidz, commonly known as djulis) is a close relative of quinoa, is rich in nutritional value and high active components, such as, betaine and polyphenols, and is a vital food crop. We investigated the effects of calcium carbonate on the growth and physiology of Djulis sprouts because calcium is an essential nutrient for plants that can strengthen a plant's root system and improve its nutrient absorption; moreover, under abiotic stress, calcium transmits messages to enhance cell tolerance. Experiments were conducted using 0, 1.25, 2.5, and 5 mM calcium carbonate hydroponic liquid treatment. Treatment with 5 mM calcium carbonate promoted the growth of djulis; specifically, root length, plant height, aboveground fresh weight, and dry weight increased by 55%, 12%, 37%, and 17%, respectively. In further investigation of the physiological change of djulis sprouts treated with calcium carbonate, the results showed that after 5 days of treatment with 5 mM calcium carbonate, the contents of hydrogen peroxide and malondialdehyde decreased significantly while the chlorophyll content increased significantly. Antioxidant enzyme activity was significantly improved. The activities of superoxide dismutase, catalase, ascorbate peroxidase, and glutathione reductase were increased by 42%, 25%, 35.7%, and 56.4%, respectively, and the calcium content of the aboveground and underground plant parts was increased fourfold and threefold, respectively. The content of proline, regarded as an osmoprotectant, was reduced by 12%. Thus, we concluded that treatment of djulis sprouts with 5 mM calcium carbonate can improve their antioxidant capacity, reduce the content of reactive oxygen species, and promote crop growth.

**Keywords:** antioxidant enzymes; calcium carbonate; calcium ion; djulis; osmotic adjustment; potassium ion

## 1. Introduction

Djulis (Chenopodium formosanum Koidz.) is native to Taiwan and has an affinitive relationship with Chenopodium quinoa Willd. Djulis is a shallow-rooted plant with spike-shaped inflorescences and is capable of producing grain at its main and axillary shoots. Because djulis grains are similar to those of gramineous plants, djulis is classified as a pseudocereal. The leaves and seeds are the parts of djulis commonly consumed. Djulis seeds are gluten free and contain abundant protein, essential amino acids, beneficial unsaturated fatty acids (e.g., linoleic acid), various minerals (e.g., Ca, P, Fe, Cu, and Zn), and nutrients (e.g., vitamins A, B2, C, and E). The seeds also contain rare elements (e.g., Se and Ge) that strengthen the immune system. Therefore, djulis consumption provides many benefits to the human body [1], and the nutritional value of djulis has gradually begun to attract research attention.

Quinoa is a dicotyledonous annual plant in the family Amaranthaceae and is highly tolerant to frost damage, salt damage, and drought [2]. Quinoa is rich in protein, lipids, fiber, and other major nutrients [3]. Therefore, the United States National Aeronautics and Space Administration routinely sends quinoa with astronauts on space missions to ensure

their vitality [4–6]. Djulis (C. formosanum Koidz.) is an herbaceous dicotyledonous annual plant belonging to the Chenopodium genus and is native to Taiwan. The seeds of djulis contain numerous elements necessary to the human body, including amino acids, minerals, starch, and protein; in particular, their potassium, phosphorus, calcium, and magnesium contents are higher than those of other cereal crops [1]. Moreover, djulis sprouts can grow under high salt stress [7]. Djulis is thus a valuable food crop.

The mineral calcium is an essential nutrient for plants. It is mainly distributed in cell organelles such as the cell wall and endoplasmic reticulum, and it plays a major role in regulating plant growth and development [8]. To increase the growth potential of crops, exogenous calcium is often applied to crops during the growing period. When common soybeans are treated with increasing concentrations of calcium, the dry weight of soybean stems, leaves, and roots and the grain yield correspondingly increase [9]. In rice, applying exogenous calcium before heading can increase the carbohydrate content in flag leaves by 15% and transport it to grains for storage [10]. Furthermore, applying exogenous calcium to fruits and vegetables after harvest can extend their storage time and maintain their quality [11,12]. Treatment with exogenous calcium can also improve crop resistance to insects. However, when applying exogenous calcium, the treatment concentration must be carefully controlled because calcium has an antagonistic relationship with potassium. If the calcium concentration is too high, it often reduces the potassium content in crops, affecting yield and quality [13].

The content of calcium ions in plant cells is related to plant adaptation to external environmental stimuli [14]. For crops grown under stressful conditions such as high temperature [15], drought [16] and high salinity [17], treating them with calcium increases the activity of antioxidant enzymes, such as superoxide dismutase (SOD), catalase (CAT), ascorbate peroxidase (APX), and glutathione reductase (GR), which can reduce the content of active oxygen and improve the stress tolerance of the crops. Calcium treatment reduces the damage to crops when they are grown in an environment in which heavy metals such as cadmium [18], nickel [19], or arsenic [20] are present. Moreover, treating crops grown under stressful conditions can increase the content of proline, increase the concentration of osmoprotectants, maintain cell membrane integrity, and improve crop tolerance to various stressors [17,18].

During the growth period of Djulis will be stunted due to stresses. Especially, the djulis sprouts are particularly sensitive to environmental changes. Therefore, the focus of this study is to improve stress tolerance during seedling growth. Because few studies have investigated the effects of exogenous calcium treatment on the growth and physiological response of quinoa or djulis, this study sought to understand the effect of $CaCO_3$ on the growth and physiological changes of djulis sprouts.

## 2. Materials and Methods

### 2.1. Reagents

β-Nicotinamide adenine dinucleotide 2′-phosphate reduced tetrasodium salt hydrate (NADPH), Diethanolamine, L-Ascorbic acid (AsA), L-Glutathione oxidized (GSH), Nicotinamide adenine dinucleotide (NADH), Magnesium chloride anhydrous, Thiobarbituric acid, Titanium chloride (TBA), Triethanolamine hydrochloride, Trizma hydrochloride were purchased from Sigma Chemical Co. (St. Louis, MO, USA). β—mercaptoethanol was purchased from bioWORLD (Dublin, OH, USA). Triethanolamine hydrochloride was purchased from Alfa Aesar (Haverhill, MA, USA). Anhydrous Hydroxylammonium Chloride, Calcium carbonate, Perchloric acid, Potassium Phosphate Dibasic, Sodium Phosphate Dibasic, Sulfosalicylic acid, Sulfuric acid were purchased from Nihon Shiyaku Reagent (Nihon Shiyaku Industries, Taipei City, Taiwan). Acetic acid, EtOH, Hydrogen Peroxide, Nitric acid, Toluene were purchased from Union Chemical Works LTD. (Kaohsiung, Taiwan), Ninhydrin was obtained from koch-light laboratories Ltd. (Haverhill, MA, USA). Ethylen ediaminetetraacetic Acid (EDAT), Manganese(II) Chloride 4-hydrate, Trichloroacetic acid

(TCA) were purchased from PanReac AppliChem(Darmstadt, Germany). All chemicals were of reagent grade and were used without further purification.

### 2.2. Plant Material and Growth

After being soaked in distilled water for 1 h, the seeds of djulis were placed in a petri dish to accelerate their germination. Subsequently, they were grown hydroponically, and the nutrient solution, defined as Yoshida nutrient solution, was formulated according to the crop testing method proposed by Yoshida et al. [21]. During crop growth, the hydroponic liquid will be replaced every 3 days to stabilize the pH value of the nutrient solutions. The djulis plants were grown at 26 °C and were illuminated by a 30,000 lux fluorescent light for 24 h. After the first leaf was fully expanded, the plants were moved to a controlled environment greenhouse to grow for 1 wk. The temperature of the greenhouse was controlled at 25–30 °C, humidity at 70–80% and the lighting was natural light. approximately 62,000–45,000 lux.

### 2.3. Calcium Treatment

After the djulis had grown for 1 wk, they were variously treated for approximately 3–5 d with Yoshida nutrient solutions containing 0, 1.25, 2.5, and 5 mM $CaCO_3$ and the resulting plant heights, fresh weights, and dry weights were measured. Ten djulis plants were collected from each treatment for measurement, and this process was repeated three times. Subsequently, plant material relevant to physiological growth measurement was taken from djulis sprouts treated with 5 mM $CaCO_3$ for 5 d for analysis.

### 2.4. Physiological Parameter Assays

The chlorophyll assay was performed using the method proposed by Wintermans and De Mots [22]. The first pair of leaves (including the terminal bud) of a djulis was collected and ground in an ice bath of 2 mL of sodium phosphate buffer (50 mM, pH 6.8). Subsequently, 40 μL of sample homogenate was collected and put into a centrifuge tube; then, 960 μL of ethanol (EtOH 95%) was added and mixed evenly with the sample homogenate. The mixture was placed in darkness for 30 min and then centrifuged at $744\times g$ at 4 °C for 15 min. Finally, take the supernatant and measure it with a spectrophotometer (Hitachi, U-2900) to determine the content of chlorophyll at 665- and 649-nm wavelengths. The malondialdehyde (MDA) assay was conducted according to the method proposed by Heath and Packer [23]. The first pair of leaves counted from the terminal bud (including the terminal bud) of the seedling was collected and added into 4 mL of 5% ($w/v$) trichloroacetic acid (TCA) for grinding. The mixture was centrifuged at $7245\times g$ at 20 °C for 5 min, and 1 mL of supernatant was mixed with 4 mL of 0.5% ($w/v$) thiobarbituric acid (TBA) in 20% ($w/v$) TCA. Subsequently, the tube was placed in a 95 °C hot water bath for 30 min and then immediately put in ice to terminate the reaction. After bubbles were removed using an ultrasonic oscillator, the mixture was centrifuged for 10 min at $1258\times g$. Finally, the MDA content was determined using the spectrophotometer (Hitachi, U-2900) at 532- and 600-nm wavelengths.

A hydrogen peroxide assay was conducted using the method proposed by Jana and Choudhuri [24]. The first pair of leaves (including the terminal bud) of a djulis was collected and ground in an ice bath of 3 mL of sodium phosphate buffer (50 mM, pH 6.8, containing 1 mM hydroxylamine), which was followed by centrifugation at $3140\times g$ at 4 °C for 25 min. Subsequently, 2 mL of supernatant and 1 mL of 0.1% ($v/v$) titanium chloride were dissolved in 20% ($v/v$) sulfuric acid, yielding a mixture that was oscillated evenly and centrifuged at $1258\times g$ at room temperature for 15 min. The hydrogen peroxide content was finally determined using the spectrophotometer (Hitachi, U-2900) at the 410-nm wavelength.

### 2.5. Analysis of Antioxidant Enzyme Activity

The first pair of leaves (including the terminal bud) of djulis plants was collected and ground in an ice bath of 0.1 M sodium phosphate buffer (pH 6.8) and was centrifuged at

$7245\times g$ at 4 °C; the supernatant was used as the extract in subsequent analyses. SOD activity was determined using the method of Paoletti et al. [25]. Specifically, 2.73 mL of reactive mixture was employed, containing 100 mM triethanolamine–diethanolamine buffer (pH 7.4), 7.5 mM reduced nicotinamide adenine dinucleotide (NADH), ethylenediaminetetraacetic acid (EDTA)/$MnCl_2$ (100 mM/50 mM, pH 7.4), 10 mM β-mercaptoethanol, and 0.2 mL of extract. After being mixed evenly, the reactive mixture received an addition of NADH for 10 min of reaction, and absorbance was measured at the wavelength of 340 nm. In this study, one unit of SOD was defined as the enzyme activity that inhibits 50% of the NADH oxidation rate in blank samples. CAT activity was determined using the method of Kato and Shimizu [26]. The extent of $H_2O_2$ reduction was measured at 240 nm, and the extinction coefficient (40 mM$^{-1}$ cm$^{-1}$) was used to calculate the activity of CAT. One unit of CAT activity was defined as the amount of CAT required to decompose 1 mole of $H_2O_2$ per min. APX activity was determined using the method proposed by Nakano and Asada [27]. As the concentration of ascorbic acid (AsA) decreases, the absorbance at 290 nm also decreases. Thus, the extinction coefficient of AsA (2.8 mM$^{-1}$ cm$^{-1}$) can be used to calculate APX activity. One unit of APX was defined as the amount of APX required to decompose 1 mole of AsA per min. GR activity assay was performed using the method of Foster and Hess [28]. One unit of GR was defined as the amount of enzyme required to decrease 1 absorbance per min at 340 nm. The protein content of the enzyme extract was determined using the method of Bradford [29].

### 2.6. Proline Assay

The proline assay was performed according to the method proposed by Bates et al. [30]. The first pair of leaves (including terminal bud) of djulis plants was collected and ground in 5 mL of 3% (*w/v*) sulfosalicylic acid and was then centrifuged at $2608\times g$ for 20 min at room temperature. After adding 1 mL of supernatant to 1 mL of ninhydrin and 1 mL of acetic acid, the mixture tube was placed in a 100 °C hot water bath for 60 min of reaction and then immediately submerged in an ice bath to terminate the reaction. Subsequently, the mixture received the addition of 4 mL of toluene, was oscillated for 15 min, and was placed at rest for 10 min. The spectrophotometer (Hitachi, U-2900) was used to determine the proline content at 520 nm.

### 2.7. Assay of Na$^+$ and K$^+$

The djulis plants were divided into aboveground and underground parts. After drying both categories of parts in an oven, 0.5 g of each part was collected and placed into a glass decomposition tube, into which 5 mL of diacid mixture [HNO3:HClO4 = 4:1 (*v/v*)] was added. The mixture was left overnight and then heated in a decomposing furnace; it was first heated at 65 °C for 15 min and then at 100 °C until the gases in the tube turned nearly transparent. The mixture was then heated to 190 °C to ensure complete volatilization of acid gases. The process was completed when the fluid at the bottom of the tube became almost transparent. After the tube was cooled, the fluid was filtered using filter paper (Whatman 42), and the sample was quantified to 50 mL using double-distilled water. Finally, the Na$^+$ and K$^+$ contents were determined using the model 410 flame photometer (Sherwood Scientific Ltd., Cambridge, UK).

### 2.8. Statistical Analysis

All the assays were performed on the basis of completely randomized design. SAS 9.4 (SAS Institute Inc., Cary, NC, USA) was employed to calculate the least significant difference to determine the differences between various treatments ($p \leq 0.05$).

## 3. Results

### 3.1. Establishment of Conditions for the Treatment of Djulis Sprouts with Calcium

Djulis sprouts were grown for 1 wk and then placed in hydroponic solution variously supplemented with 0, 1.25, 2.5, and 5 mM $CaCO_3$ for 7 d, after which the root length, plant

height, and dry and fresh weights of above- and belowground parts of the sprouts were measured. As evident in Table 1, for 5 mM CaCO$_3$ treated sprouts on average, the shoots length was 3.74 cm, plant height was 6.64 cm, fresh ground weight was 1.25 g, and dry ground weight was 0.18 g, all of which were higher than those of the control group. Moreover, plant appearance improved with 5 mM CaCO$_3$ treatment (Figure 1D). No significant difference was identified between the dry and fresh weights of the underground parts treated with and without CaCO$_3$ (Table 1). Thus, the results showed that 5 mM was the optimal CaCO$_3$ treatment concentration.

**Table 1.** The effect of calcium carbonate on the growth traits of Djulis sprouts. The 7-d-old djulis plants were treated with 0 mM, 1.25 mM, 2.5 Mm and 5 mM CaCO$_3$ for 7 days, the growth characteristics of seedlings were investigated. Data are means $\pm$ SE ($n = 10$), repeated three times. Values with the same letter are not significantly different at $p < 0.05$.

| CaCO$_3$ Concentration (mM) | Root Length (cm) | Shoot Height (cm) | Shoots | | Roots | |
|---|---|---|---|---|---|---|
| | | | Fresh Weigh (g) | Dry Weigh (g) | Fresh Weigh (g) | Dry Weigh (g) |
| 0 | 2.41 $\pm$ 0.11 [b] | 5.90 $\pm$ 0.04 [b] | 0.912 $\pm$ 0.01 [b] | 0.162 $\pm$ 0.002 [b] | 0.082 $\pm$ 0.024 [a] | 0.022 $\pm$ 0.001 [a] |
| 1.25 | 2.33 $\pm$ 0.37 [b] | 5.96 $\pm$ 0.3 [b] | 0.984 $\pm$ 0.11 [a,b] | 0.168 $\pm$ 0.019 [b] | 0.090 $\pm$ 0.008 [a] | 0.025 $\pm$ 0.002 [a] |
| 2.5 | 2.53 $\pm$ 0.08 [b] | 6.26 $\pm$ 0.03 [a,b] | 1.064 $\pm$ 0.11 [a,b] | 0.163 $\pm$ 0.023 [b] | 0.115 $\pm$ 0.011 [a] | 0.024 $\pm$ 0.003 [a] |
| 5 | 3.74 $\pm$ 0.24 [a] | 6.64 $\pm$ 0.1 [a] | 1.252 $\pm$ 0.05 [a] | 0.189 $\pm$ 0.014 [a] | 0.109 $\pm$ 0.033 [a] | 0.027 $\pm$ 0.006 [a] |

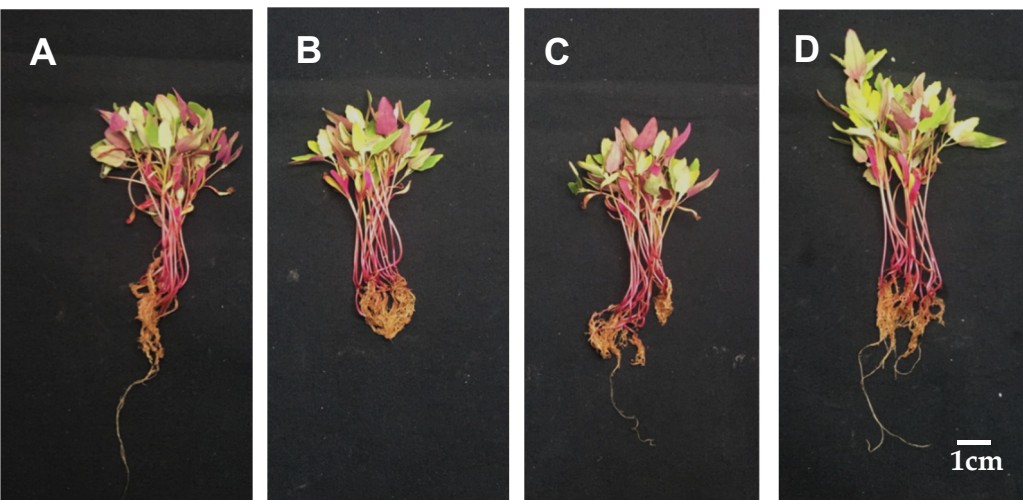

**Figure 1.** The effect of calcium carbonate on the growth of Djulis sprouts. The 7-d-old djulis plants were treated with 0 mM (**A**), 1.25 mM (**B**), 2.5 mM (**C**) and 5 mM (**D**) CaCO$_3$ for 7 days, the growth characteristics of seedlings were recorded. (Scale bar = 1.0 cm).

Growing time was assessed next. Djulis sprouts were grown for 1 wk and then placed in 5 mM CaCO$_3$ hydroponic solution for 3, 5, or 7 d. No significant difference was evident in plant height (Figure 2A), but root length, fresh weight, and dry weight of the underground parts all increased as treatment duration increased (Figure 2B–D). Thus, treating djulis sprouts with calcium for 5 d was optimal treatment time.

*3.2. Effect of Calcium on Physiology of Djulis Sprouts*

In the next experiment, djulis sprouts were treated with 5 mM CaCO$_3$ for 5 d, after which the changes in chlorophyll, H$_2$O$_2$, and MDA contents were detected. The chlorophyll content of djulis sprouts treated with CaCO$_3$ was significantly higher than that of the control group (Figure 3A), whereas the contents of H$_2$O$_2$ and MDA were significantly lower than those of the control group (23.5% and 23.7% reductions, respectively; Figure 3B,C). Thus,

treating djulis sprouts with calcium can reduce the content of $H_2O_2$ and MDA, thereby increasing the chlorophyll content.

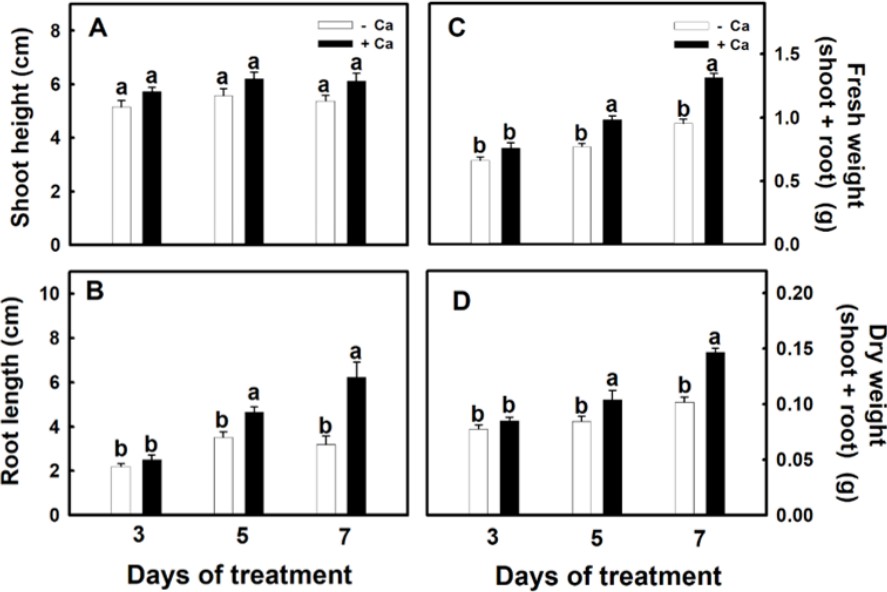

**Figure 2.** The effect of calcium carbonate on the growth characters of Djulis sprouts. The 7-d-old djulis plants were treated with 5 mM $CaCO_3$ for 3, 5, and 7 days. Shoot height (**A**), root length (**B**), root fresh weight (**C**), and root dry weight (**D**) were investigated. Bars show means ± SE (*n* = 10), repeated three times. Values with the same letter are not significantly different at *p* < 0.05.

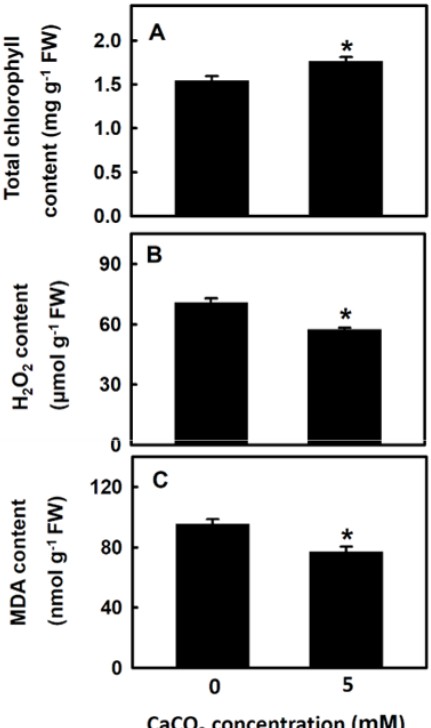

**Figure 3.** The effect of calcium carbonate on the physiological indictors of Djulis sprouts. The 7-d-old djulis plants were treated with 5 mM $CaCO_3$ for 5 days. The total chlorophyll content (**A**) $H_2O_2$ content (**B**) and MDA content (**C**) of djulis were measured. Bars show means ± SE (*n* = 4). * represent values that are significantly different between calcium carbonate concentration 0 mM and 5 mM at *p* < 0.05.

### 3.3. Effect of Calcium on Antioxidant Enzyme Activity of Djulis Sprouts

The next experiment assessed whether calcium treatment affects antioxidant enzyme activity in djulis sprouts. The activity of the antioxidant enzymes SOD, CAT, APX, and GR was significantly higher in djulis sprouts treated with 5 mM $CaCO_3$ for 5 d than in the untreated group, with increases of approximately 42%, 25%, 35.7%, and 56.4%, respectively (Figure 4). Thus, the treatment of djulis sprouts with calcium can increase antioxidant enzyme activity.

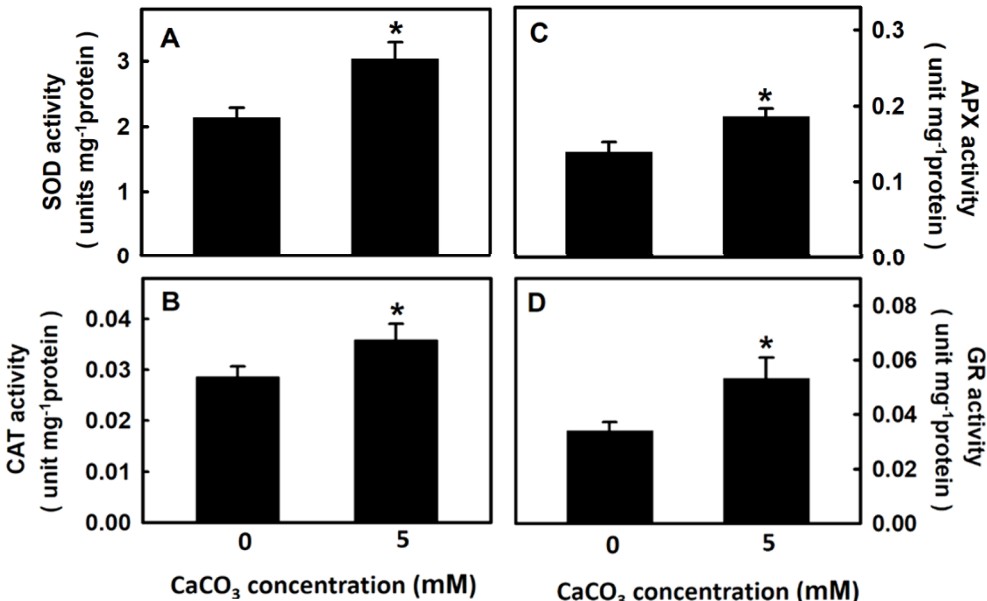

**Figure 4.** The effect of calcium carbonate on antioxidant enzyme activities of Djulis sprouts. The 7-d-old djulis plants were treated with 5 mM $CaCO_3$ for 5 days. The SOD (**A**) CAT (**B**) APX (**C**) and GR (**D**) activities of djulis were measured. Bars show means ± SE (*n* = 4). * represent values that are significantly different between calcium carbonate concentration 0 mM and 5 mM at *p* < 0.05.

### 3.4. Effect of Calcium on Potassium Content of Djulis Sprouts

Potassium and calcium ions are antagonistic to each other, and thus, this study investigated whether treating djulis sprouts with $CaCO_3$ would affect the potassium ion content. The results revealed that after treatment with 5 mM $CaCO_3$ for 5 d, the content of calcium ions in both the aboveground and underground parts of the djulis sprouts increased significantly (Figure 5A,C). For the aboveground parts, no significant difference in potassium ion content was evident between the treatment and control groups, but the potassium ion content in the lower underground parts of the djulis sprouts was significantly lower than that of the control groups.

### 3.5. Effect of Calcium on Proline Content of Djulis Sprouts

In plants, proline is an osmoprotectant and an indicator of the degree of stress damage. This study revealed that the proline content of djulis sprouts treated with 5 mM $CaCO_3$ for 5 d was significantly lower than that of the control groups by approximately 12% (Figure 6). Therefore, calcium treatment of djulis sprouts may reduce stress damage.

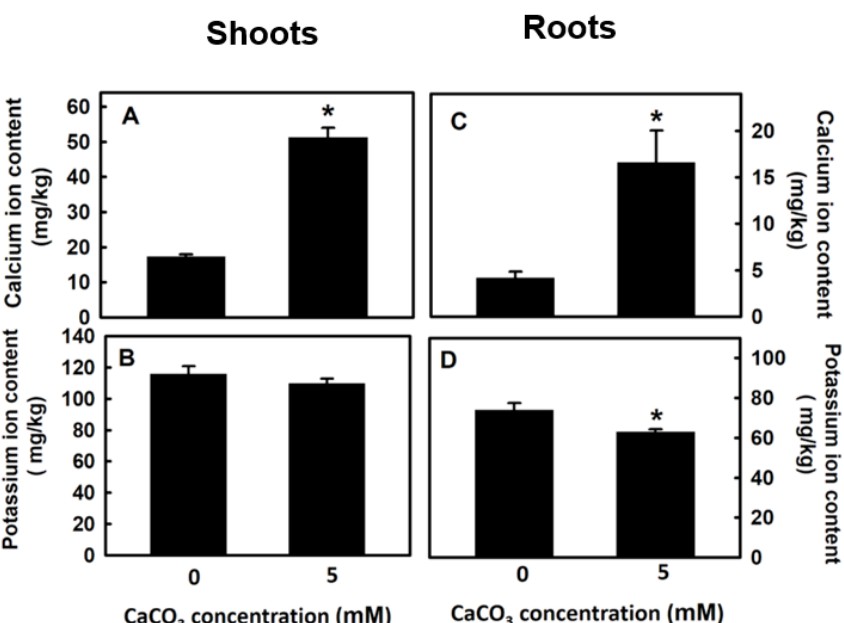

**Figure 5.** The effect of calcium carbonate on $Na^+$ and $K^+$ content of Djulis sprouts. The 7-d-old djulis plants were treated with 5 mM $CaCO_3$ for 5 days. The $Ca^+$ content of the shoot (**A**) and the root (**C**), and $K^+$ content of the shoot (**B**) and the root (**D**) of djulis were measured. Bars show means $\pm$ SE (*n* = 4). * represent values that are significantly different between calcium carbonate concentration 0 mM and 5 mM at $p < 0.05$.

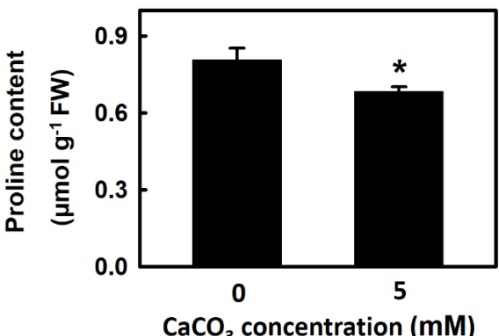

**Figure 6.** The effect of calcium carbonate on proline content of Djulis sprouts. The 7-d-old djulis plants were treated with 5 mM $CaCO_3$ for 5 days. The proline content of djulis were measured. Bars show means $\pm$ SE (*n* = 4). * represent values that are significantly different between calcium carbonate concentration 0 mM and 5 mM at $p < 0.05$.

## 4. Discussion

Calcium is an essential element for plant growth and development. Its functions are to maintain the stability of cell membranes, constitute the cell wall, play a critical role in the transmission of information under stressful conditions, and regulate physiological responses to mitigate injuries [8]. A lack of calcium during plant growth and development negatively affects yield and quality [31]. Therefore, exogenous calcium is often applied during growth to optimize crop yield and quality. Treating strawberries with 0.5% calcium chloride can significantly increase the number of leaves per plant, leaf area, and chlorophyll content of leaves in the vegetative growth period and significantly increase the number of flowers per plant, fruit weight, and fruit weight per plant in the reproductive growth period [32]. Treating common soybeans with increasing calcium concentrations (1.10, 1.65, 2.20, 2.75, and 3.30 mmol L−1) increases the dry weight of soybean stems, leaves, and roots

and the fresh weight of soybean seeds [9]. Administering $CaCO_3$ by spraying leaves during rice growth improves rice yield and quality and enhances rice resistance to insects [33].

In this study, djulis sprouts were treated with various concentrations of $CaCO_3$. After treatment with 5 mM $CaCO_3$, the calcium content of aboveground and underground parts of djulis sprouts increased threefold and fourfold, respectively (Figure 5A,C) Moreover, An average of the root length, plant height, fresh weight, and dry weight of djulis sprouts increased by 55%, 12%, 37%, and 17%, respectively (Table 1). Furthermore, the appearance of the plant improved with 5 mM $CaCO_3$ treatment (Figure 1D). Thus, exogenous calcium application promotes the growth of djulis sprouts.

Hydrogen peroxide ($H_2O_2$) is produced predominantly in plant cells during photosynthesis and respiration. This substance is the most stable in the active oxygen family and participates in the regulation of plant growth and physiological reactions [34]. $H_2O_2$ interacts with calcium and regulates plant growth in response to the environment [35]. In this study, treatment with $CaCO_3$ increased the calcium content of both the aboveground and underground parts of djulis sprouts (Figure 5A,C) and also increased the dry and fresh weight of the aboveground parts (Table 1). $H_2O_2$ may also be involved in the regulatory response of calcium because $H_2O_2$ has been considered a molecule by which plants transmit messages when under stress [36].

Antioxidant defense mechanisms of plants are activated when a plant is under stress, and these antioxidants participate in maintaining the structural integrity of the cell components and alleviating oxidative damage. The antioxidant defense system consists of antioxidants such as Ascorbate and glutathione and antioxidant enzymes such as SOD, CAT, APX, and GR. When $H_2O_2$ levels are reduced due to antioxidant system effects, $H_2O_2$ is considered an important messenger molecule for regulating plant growth and development [36]. A study showed that in tobacco treated with 20 mM calcium chloride in a high-temperature environment, the activities of CAT, APX, and GR were increased, which in turn reduced the $H_2O_2$ content and increased the photosynthesis rate [15]. Moreover, where rice was treated with the heavy metal arsenic to inhibit growth and reduce biomass, additional simultaneous treatment with 10 mM calcium chloride increased the activity of the enzymes SOD, CAT, and APX, thereby reducing the $H_2O_2$ and MDA content and mitigating the harm of arsenic [20]. In this study, after treatment with 5 mM $CaCO_3$, the enzyme activity of SOD, CAT, APX, and GR increased in djulis sprouts (Figure 4), $H_2O_2$ and MDA contents were reduced (Figure 3B,C), and chlorophyll content was thus increased (Figure 3A). Thus, exogenous calcium treatment can improve antioxidant enzyme activity, which not only reduce the content of hydrogen peroxide, but also decreases lipid peroxidation, thereby stabilizing cell membrane structure and maintain the integrity of the chloroplast.

Proline is an osmoprotectant. When quinoa grows under stressful conditions, the content of proline increases to improve its resistance to adversity [37]. Under drought stress conditions and compared with control groups, quinoa treated with exogenous proline exhibited increased plant height, freshness, and dry weight; moreover, proline was also shown to increase the content of both carbohydrate and protein in quinoa seeds [38]. Treatment with cadmium adversely affects mustard height, root length, and dry weight. If 50 mM calcium chloride and heavy metal cadmium are simultaneously applied to mustard, the activity of the antioxidant enzymes SOD, APX, and GR is increased as well as proline content, which reduces the damage from cadmium [18]. An inference to draw from this may be that for plants grown under stressful conditions, crop tolerance to stress can be increased through exogenous calcium treatment because of its positive effect on proline content. In the present study, after treatment with 5 mM $CaCO_3$, djulis sprouts exhibited increased antioxidant enzyme activity (Figure 4) and reduced $H_2O_2$ content (Figure 3B). The reduced $H_2O_2$ content means that oxidative stress condition was diminishing, thus proline content also decreased (Figure 6).

During the plant growth period, exogenous calcium application stabilizes crop yields and quality, but in crops, calcium and potassium have an antagonistic relationship. Thus,

if the concentration of calcium is too high, it may affect potassium absorption and reduce yield and quality [13]. In this study, djulis sprouts were treated with 5 mM CaCO$_3$. Both the calcium content of the aboveground and underground plant parts significantly increased (Figure 5A,C), and the potassium content of the aboveground parts was not affected; however, the potassium content of the underground parts was significantly reduced (Figure 5B,D). Yet, this did not affect the growth of the underground parts of djulis sprouts (Table 1).

## 5. Conclusions

In this study, treating djulis sprouts with CaCO$_3$ increased antioxidant enzyme activity, which in turn reduced H$_2$O$_2$ content. A tendency to decrease the content of proline is induced. Therefore, the decrease both in H$_2$O$_2$ and MDA content probably display that reduce lipid peroxidation to maintain chloroplast membrane structure, increase chlorophyll content and ultimately promoted the growth of the shoots of djulis sprouts (Figure 7).

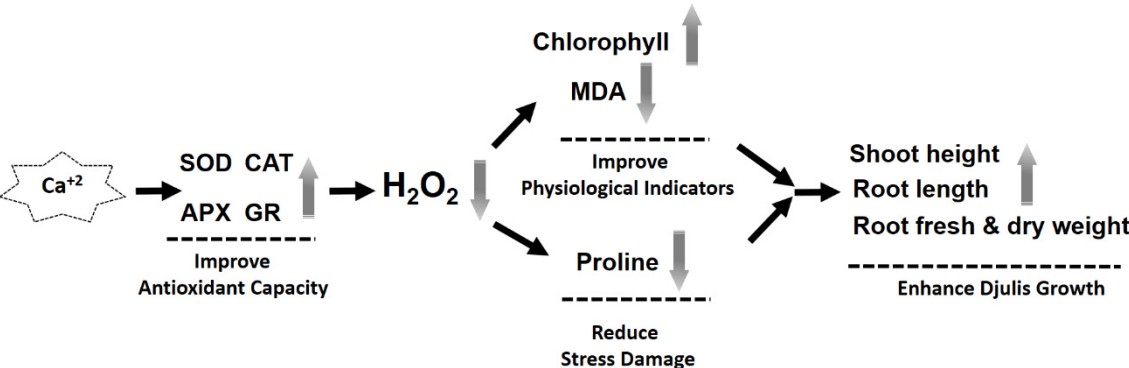

**Figure 7.** Mechanism diagram of calcium carbonate to promote the growth of djulis sprouts.

**Author Contributions:** Conceptualization, Y.-Y.C.; methodology, formal analysis and investigation, W.-J.W. and Y.-T.L.; writing—original draft preparation, Y.-Y.C.; writing—review and editing, Y.-Y.C. All authors have read and agreed to the published version of the manuscript.

**Funding:** This research received no external funding.

**Institutional Review Board Statement:** Not applicable.

**Informed Consent Statement:** Not applicable.

**Data Availability Statement:** Data sharing not applicable.

**Conflicts of Interest:** The authors declare no conflict of interest.

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
