# Peer review of "Effect of Calcium on the Growth of Djulis (Chenopodium formosanum Koidz.) Sprouts"

_agronomy, doi:10.3390/agronomy11010082_

Round 1

Reviewer 1 Report

The manuscript presented by Chao et al. describes the effect of calcium supply on the growth of Djulis sprouts. The authors thereby first determined the optimal concentration of calcium carbonate supply and subsequently measured physiological parameters such as chlorophyll content and antioxidant enzyme activity. The manuscript is well-written, methods are explained in detail and results presented in an adequate way. I have however several questions/suggestions, which I think should be answered before the manuscript could be accepted for publication:

  1. In the first paragraph of the introduction, the authors highlight in detail the beneficial effects of Djulis seeds on human nutrition, however, it seems that all analyses have been performed on sprouts. Do the authors have data on grain yield? Can they explain which parts of the plants presented in the results are used for human consumption or in general explain better why sprouts have been used?

  1. In line 157 it says “for 7 d” while the legend for figure 1 says “for 3 days”. Please clarify.

  1. There are some discrepancies in the results of table 1 and figure 2 which need further explanation:

- In table 1 the root length is significantly increased at 5 mM CaCo3, however the fresh and dry weight of the roots shows no significant increase. Can the authors explain why the increase in root length is not reflected in an increase in weight?

- Table 1 shows a significant increase in shoot height and root length at 3 days, however there is no significant increase for these parameters in Figure 2A and B. Please explain.

- Root fresh weight and dry weight in table 1 lie between 0.08 and 0.1 g and around 0.02 g, respectively. In figure 2C and D these parameters have values around 1 mg or 0.1 mg, respectively. Please explain the difference.

  1. In Figures 3, 4, 5 and 6 the legend says “n=4”. Please explain what n is: plants, experiments, technical repeats… Please also state how many times the experiment was repeated with similar results.

  1. Lines 344 to 367 indicate that the reduction in H2O2 has a positive effect on plant growth, chlorophyll content and membrane stability, but it does not entirely become clear how. Please explain the connection better.

  1. Conclusions: The conclusions are too strong, for example it says “…reduced H2O2 content, subsequently reducing proline content.” There is no direct evidence that the reduction in H2O2 is the cause for the reduction in proline, please write “probably” or something similar. Also “…was demonstrated to lead to an increase in cell membrane stability, which in turn improved…”, again no direct evidence, please write in a more suggestive way.

Minor points:

- line 11: “functional components”, meaning not clear

- line 25: “the proline content of the osmotic adjustment substance”, sentence not clear

- line 27-28: “stabilize the cell membrane structure”, no evidence for this in the results

- line 91: after centrifugation, what was used in the spectrophotometer, the supernatant?

- line 159: “for treated sprouts”, treated with 5mM?

- legend to Figure 1: 2.5 Mm -> 2.5 mM

- in Figure 3, panel A and B overlap

- the discussion starts with some sentences about quinoa, which is not the direct subject of the study

- line 340: “length”, root length?

Author Response

 List of response to Reviewer #1:

Point 1:

In the first paragraph of the introduction, the authors highlight in detail the beneficial effects of Djulis seeds on human nutrition, however, it seems that all analyses have been performed on sprouts. Do the authors have data on grain yield? Can they explain which parts of the plants presented in the results are used for human consumption or in general explain better why sprouts have been used?

Response 1: 

In this experiment, Djulis sprouts was used as the experimental material to explore whether external application of calcium can improve the stress tolerance of seedlings. During the growth period of Djulis will be stunted due to stresses. Especially, the djulis sprouts are particularly sensitive to environmental changes. Therefore, the focus of this study is to improve stress tolerance during seedling growth period.

The sentence has been rewritten in Line 83-85.

Point 2:

In line 157 it says “for 7 d” while the legend for figure 1 says “for 3 days”. Please clarify.

Response 2: 

In line 213 it says “for 7 d” is the correct and it has been corrected in figure 1.

Point 3:

There are some discrepancies in the results of table 1 and figure 2 which need further explanation:

3.1 In table 1 the root length is significantly increased at 5 mM CaCo3, however the fresh and dry weight of the roots shows no significant increase. Can the authors explain why the increase in root length is not reflected in an increase in weight?

3.2 Table 1 shows a significant increase in shoot height and root length at 3 days, however there is no significant increase for these parameters in Figure 2A and B. Please explain.

3.3 Root fresh weight and dry weight in table 1 lie between 0.08 and 0.1 g and around 0.02 g, respectively. In figure 2C and D these parameters have values around 1 mg or 0.1 mg, respectively. Please explain the difference.

Response 3: 

3.1  Djulis seedlings can increase root length and fresh weight after calcium treatment, but dry weight does not increase significantly. In this study, hydroponic cultivation was used. It is speculated that calcium treatment will promote water absorption and then increase fresh weight. However, the result on the dry matter after drying is not obvious.

3.2  In table 1, The 7-d-old djulis plants were treated with 0 mM, 1.25 mM, 2.5 Mm and 5 mM CaCO3 for 7 days, isn’t for ”3 days”. It has been corrected in table 1. 2A and 2B are consistent with the results in Table 1

3.3 Re-check the results in Figure 3 and find that the Y-axis text and parameter units are wrong. Figure 3 has been corrected. The corrected content is that the unit of the parameter is “g”, not “mg”; the Y-axis text of Fig 3C and 3D are the fresh weigh and dry weigh of the whole plant, respectively.

In figure 3 C. the fresh weigh of whole plants treated with or without 5 mM CaCO3 for 7 days are approximately1.3 g and 0.99g, respectively. This result is similar to Table 1. However, the results in Figure 3D and Table 1 are inconsistent, and it is speculated that it may be a difference in biological repetition.

Point 4:

In Figures 3, 4, 5 and 6 the legend says “n=4”. Please explain what n is: plants, experiments, technical repeats… Please also state how many times the experiment was repeated with similar results.

Response 4:

In Figures 3, 4, 5 and 6 the legend says “n=4”,”N” means biological repetition.

Point 5:

Lines 344 to 367 indicate that the reduction in H2O2 has a positive effect on plant growth, chlorophyll content and membrane stability, but it does not entirely become clear how. Please explain the connection better.

Response 5:

Rewrite the sentence as “exogenous calcium treatment can improve antioxidant enzyme activity, which not only reduce the content of hydrogen peroxide, but also decreases lipid peroxidation, thereby stabilizing cell membrane structure and maintain the integrity of the chloroplast.” Line 407-410.

Point 6:

Conclusions: The conclusions are too strong, for example it says “…reduced H2O2 content, subsequently reducing proline content.” There is no direct evidence that the reduction in H2O2 is the cause for the reduction in proline, please write “probably” or something similar. Also “…was demonstrated to lead to an increase in cell membrane stability, which in turn improved…”, again no direct evidence, please write in a more suggestive way.

Response 6:

Rewrite the conclusions as “In this study, treating djulis sprouts with CaCO3 increased antioxidant enzyme activity, which in turn reduced H2O2 content. A tendency to decrease the content of proline is induced. Therefore, the decrease both in H2O2 and MDA content probably display that reduce lipid peroxidation to maintain chloroplast membrane structure, increase chlorophyll content and ultimately promoted the growth of the shoots of djulis sprouts (Figure 7).” Line 434-438.

Point 7: Minor points:

7.1 - line 11: “functional components”, meaning not clear

7.2 - line 25: “the proline content of the osmotic adjustment substance”, sentence not clear

7.3 - line 27-28: “stabilize the cell membrane structure”, no evidence for this in the results

7.4- line 91: after centrifugation, what was used in the spectrophotometer, the supernatant?

7.5- line 159: “for treated sprouts”, treated with 5mM?

7.6- legend to Figure 1: 2.5 Mm -> 2.5 Mm

7.7- in Figure 3, panel A and B overlap

7.8- the discussion starts with some sentences about quinoa, which is not the direct subject of the study

7.9- line 340: “length”, root length?

Response 7:

7.1 Rewrite” functional components” to ”high active components, such as, betaine and polyphenols.” Line 11-12.

7.2 Revise to “The content of Proline, regarded as an osmoprotectant, was reduced by 18%.” Line 28-29.

7.3 The stabilize the cell membrane structure be deleted

7.4 Revise to “Take the supernatant and measure it with a spectrophotometer (Hitachi, U-2900) to determine the content of chlorophyll at 665- and 649-nm wavelengths.” Line 126

7.5 Revise to” for 5 mM CaCO3 .” Line 196.

7.6 Corrected. Line 213.

7.7 Corrected. Line 265.

7.8 The first paragraphs in the discussion has been placed in the introduction.  Please check line 51-59.

7.9 Corrected. Line 382.

Reviewer 2 Report

Comment on Ms: agronomy-1026329.

 This manuscript describes the positive effects by Ca addition on the physiological parameters of Djulis (Chenopodium formosan), such as seedling growth, height, fresh weights and dry weights of shoot. The results also show increased chlorophyll content and antioxidants enzyme activities, but decrease of MDA and H2O2. Such effects were shown in several investigations before with other plant species, so the results are not quite new.

The results and figures are mainly easy to read and understand, but the cultivation method needs additional information, and also the statistics. The first paragraphs in the Discussion would be better placed in the Introduction.

Materials and Methods

 Was the pH of the solution constant during cultivation? Addition of CaCO3 should increase pH if no adjustment was done. What temperature was the “room temperature”?

What was the humidity during cultivation? What nutrients except for Ca were used during cultivation?

How was the light condition in the green house?

Please, add company and city for all chemical substances!

Please, add company, city for the flame photometer used!

Statistics

How many biological tests (repetitions) were made?

Minor mistakes

  1. P. 2, line 44. The cytoplasm is not considered as a cell organelle.
  2. P. 3. Line 96, line 99, line 108 and so on… please use x g instead of rpm.
  3. P. 4. Line 159-161. This is not correct. These data represent the fresh and dry weights of shoots, not roots.

Fig. 2. The time 3, 5 and 7 days mentioned here might be better explained! I think you mean that experiments were performed when the plants were 7 + 3, 7 + 5 and 7 + 7 day-old?

Fig. 3. On the Y-axis the text for A and B are almost overlapping each other.

  1. 9. Line 354. Please, explain AsA!
  2. 10. Line 381. …Stabilizes crop yield.

Author Response

List of response to Reviewer #2:

 Point 1:

The first paragraphs in the Discussion would be better placed in the Introduction.

Response 1: 

The first paragraphs in the discussion has been placed in the introduction.  Please check line 43~51.

Point 2:

2.1 Was the pH of the solution constant during cultivation? Addition of CaCOshould increase pH if no adjustment was done.

2.2 What temperature was the “room temperature”?

2.3 What was the humidity during cultivation?

2.4 What nutrients except for Ca were used during cultivation?

2.5 How was the light condition in the green house?

Response 2:

2.1 During crop growth, the hydroponic liquid will be replaced every 3 days to stabilize the pH value of the nutrients solutions. Line 98-99.

2.2 room temperature was 26 ºC. Line 99.

2.3 humidity at 70%-80%. Line 102.

2.4 After the djulis had grown for 1 wk, they were variously treated for approximately 3–5 d with Yoshida nutrient solutions containing 0, 1.25, 2.5, and 5 mM CaCO3. Line 106.

2.5 The lighting was natural light. approximately 62,000-45,000 lux. Line 103

The content of the above material method has been rewritten, Line 95-110.

Point 3:

3.1 Please, add company and city for all chemical substances!

3.2 Please, add company, city for the flame photometer used!

Response 3:

3.1, All chemical substances have been added companies and cities, and detailed in the material method, 2.1 Reagents. Line 79-93.

3.2 Information such as company and city added. Line 192-193.

Point 4:

How many biological tests (repetitions) were made?

Response 4:

Ten djulis plants were collected from each treatment for measurement, and this process was repeated three times. Please check line 107-108.

Point 5: Minor mistakes

5.1  P. 2, line 44. The cytoplasm is not considered as a cell organelle.

5.2  P. 3. Line 96, line 99, line 108 and so on… please use x instead of rpm.

5.3  P. 4. Line 159-161. This is not correct. These data represent the fresh and dry weights of shoots, not roots.

5.4  Fig. 2. The time 3, 5 and 7 days mentioned here might be better explained! I think you mean that experiments were performed when the plants were 7 + 3, 7 + 5 and 7 + 7 day-old?

5.5  Fig. 3. On the Y-axis the text for A and B are almost overlapping each other.

5.6  Line 354. Please, explain AsA!

5.7  Line 381. …Stabilizes crop yield.

Response 5:

   5.1 Cytoplasm has been deleted. Line 53

   5.2 Change all centrifugal speed units to xg.

   5.3 The roots has been changed to shoots. Line 208

5.4 In Figure 2 is mentioned that 3, 5, and 7 days refer to the days of treatment, not the seedling age. The days in the picture have been changed to days of treatment. Line 251

5.5 The revision has been completed in accordance with the comments in Fig 3.

5.6  Changed AsA to ascorbate. Line 411

5.7  Changed “stabilizes crop yield” to “stabilizes crop yields”. Line 457.
